# Overcoming barriers to engagement and adherence to a home-based physical activity intervention for patients with heart failure: a qualitative focus group study

Nduka C Okwose,[1] Nicola O'Brien,[2] Sarah Charman,[1] Sophie Cassidy,[1] David Brodie,[3] Kristian Bailey,[4] Guy A MacGowan ,[4,5] Djordje G Jakovljevic,[1,6] Leah Avery [1,7]

For numbered affiliations see end of article.

**Correspondence to**
Dr Leah Avery;
leah.avery@tees.ac.uk

## ABSTRACT

**Objectives** Clinical guidelines recommend regular physical activity for patients with heart failure to improve functional capacity and symptoms and to reduce hospitalisation. Cardiac rehabilitation programmes have demonstrated success in this regard; however, uptake and adherence are suboptimal. Home-based physical activity programmes have gained popularity to address these issues, although it is acknowledged that their ability to provide personalised support will impact on their effectiveness. This study aimed to identify barriers and facilitators to engagement and adherence to a home-based physical activity programme, and to identify ways in which it could be integrated into the care pathway for patients with heart failure.

**Design** A qualitative focus group study was conducted. Data were analysed using thematic analysis.

**Participants** A purposive sample of 16 patients, 82% male, aged 68±7 years, with heart failure duration of 10±9 years were recruited.

**Intervention** A 12-week behavioural intervention targeting physical activity was delivered once per week by telephone.

**Results** Ten main themes were generated that provided a comprehensive overview of the active ingredients of the intervention in terms of engagement and adherence. Fear of undertaking physical activity was reported to be a significant barrier to engagement. Influences of family members were both barriers and facilitators to engagement and adherence. Facilitators included endorsement of the intervention by clinicians knowledgeable about physical activity in the context of heart failure; ongoing support and personalised feedback from team members, including tailoring to meet individual needs, overcome barriers and increase confidence.

**Conclusions** Endorsement of interventions by clinicians to reduce patients' fear of undertaking physical activity and individual tailoring to overcome barriers are necessary for long-term adherence. Encouraging family members to attend consultations to address misconceptions and fear about the contraindications of physical activity in

### Strengths and limitations of this study

► Qualitative focus groups are an appropriate methodological approach to explore views on uptake, engagement and adherence to an intervention and can help facilitate consensus.

► Adults with heart failure who had declined participation in the intervention and those who had previously attended and declined participation in centre-based programmes were recruited to obtain representativeness of views.

► A higher proportion of men participated; therefore, the views of women should be explored further.

► A wide range of patients with heart failure in terms of length of time since diagnosis were recruited to the study, highlighting the broad appeal of the intervention.

the context of heart failure should be considered for adherence, and peer-support for long-term maintenance.
**Trial registration number** NCT03677271.

## INTRODUCTION

Increasing levels of physical activity is recommended to improve cardiometabolic outcomes and to enhance overall health and psychological well-being.[1–3] In the UK, cardiac rehabilitation is offered as a pathway of referral for people living with chronic conditions such as cardiovascular disease. Encouraging patients to attend is often challenging, although uptake of cardiac rehabilitation by patients following a cardiac event or procedure (eg, myocardial infarction, percutaneous coronary intervention) increased from 45% in 2013 to 50% in 2016.[4] However, these figures provide an artificial estimate of attendance in the context of heart failure, because only 5.3% of patients with heart

failure actually attend cardiac rehabilitation.[4] Furthermore, dropout rates often exceed 20% for those attending centre-based programmes.[5] Barriers to uptake, engagement and adherence are reported to include proximity of cardiac rehabilitation services and motivation levels of patients to attend.[5 6]

A potential solution to overcome these barriers is the provision of home-based physical activity programmes. These have shown to elicit similar outcomes to centre-based programmes and could be particularly attractive to those with a preference to increase activity levels at home. They also have the potential to be tailored to individual needs and preferences, thereby increasing engagement and adherence and reducing dropout rates.[7 8]

This qualitative study explored the barriers and facilitators to engagement and adherence to a home-based physical activity intervention for patients with heart failure, called 'Active-at-Home-HF'. The findings will be used to inform optimisation of the intervention for delivery by clinicians during routine care delivery.

## METHODS

All participants provided informed written consent prior to participation.

### Intervention

Active-at-Home-HF is a behavioural intervention designed to support patients with heart failure to increase their baseline physical activity levels by at least 2000 steps per day over a period of 12 weeks.[9] The intervention was developed as an alternative to centre-based cardiac rehabilitation programmes to help overcome some of the well-documented barriers to engagement and adherence (eg, travel, cost, group-based), but also as a means to support those who have previously attended centre-based programmes who wish to continue physical activity at home. Three researchers with backgrounds in exercise physiology were trained by a chartered health psychologist (LA) to deliver the intervention adopting the role of a lifestyle coach. Coaches used evidence-based behaviour change techniques, including behavioural goal setting, action planning, problem solving, self-monitoring, feedback on behaviour and review of behavioural goals.[10–12] Participants were supported to increase and sustain physical activity levels via telephone calls lasting between 8 and 15 min on at least 10 occasions, that is, weekly over a period of 12 weeks. The number of telephone calls was dictated by the participant up to a maximum of 12 calls (ie, one call per week). Physical activity goals were set and modified in collaboration with the coach and the participant depending on each participant's progress, capability, needs and preferences, with an emphasis on duration of physical activity rather than intensity. A standardised proforma was used to structure calls and to serve as a prompt to use all behaviour change techniques, where appropriate, and to increase fidelity of delivery. All three researchers who delivered the intervention also

**Table 1** Patients' demographic and clinical characteristics (mean±SD)

| Parameter | |
|---|---|
| Age (years) | 67±5 |
| Men/women | 13/3 |
| Weight (kg) | 85.2±15.9 |
| Height (m) | 1.73±0.1 |
| Duration of heart failure (years) | 10±9 |
| Medication | |
| ACE inhibitors | 12 |
| B-blockers | 16 |
| ARBs | 3 |
| Diuretics | 9 |
| Warfarin | 5 |
| ICD/pacemakers | 10 |
| Comorbidities | |
| COPD | 1 |
| Type 2 diabetes | 5 |
| Hypertension | 16 |
| Depression | 2 |
| Arthritis | 1 |

ARB, angiotensin receptor blockers; COPD, chronic obstructive pulmonary disease; ICD, Implantable cardioverter defibrillator.

completed study visits with participants during the Active-at-Home-HF pilot study.[13]

### Design

A qualitative study embedded within a pilot study of the Active-at-Home-HF intervention was conducted.[13] The aim was to deliver the intervention to participants and obtain feedback on specific aspects of it (eg, mode of delivery), but primarily to obtain views on whether an intervention of this kind could provide the level and type of support required to increase and maintain physical activity levels. Focus group discussions were used to identify barriers and facilitators to engagement and adherence, and ways in which the intervention could be integrated into routine clinical care.

### Participants

Eligible participants had left ventricular ejection fraction ≤40% diagnosed at least 3 months previously; were clinically stable and receiving an optimal medical treatment; had no contraindications to physical activity; and were able to walk and perform activities of daily living independently. Table 1 presents a summary of participant characteristics.

The aim was to recruit a purposive sample of between 12 and 20 participants in line with published guidance and previously published qualitative research to ensure data saturation.[13 14] Eligible participants included those who had previously taken part and those who had refused

participation in centre-based cardiac rehabilitation programmes to ensure a representation of views and experiences, and also included those who had refused participation in Active-at-Home-HF to identify barriers to uptake. Participants were contacted via email, telephone or spoken to in person to discuss participation in the study and were given an opportunity to ask questions to ensure they understood the aims of the study.

## Study procedure

Eligible participants under the care of a cardiologist at the Freeman Hospital, Newcastle upon Tyne, UK were invited to take part in the study. Focus group discussions were facilitated by two female chartered health psychologists (LA and NO'B) with expertise in health behaviour change, physical activity intervention and qualitative research methods. Neither of the researchers had met the participants prior to the focus group discussions and neither were aware of any preliminary findings of the Active-at-Home-HF pilot study. The Active-at-Home-HF behavioural intervention was developed by one of the health psychologists (LA) conducting the focus groups. The focus group discussions were conducted in a meeting room at the Royal Victoria Infirmary, Newcastle upon Tyne. Only the two researchers and the participants were present during the discussions. One of the researchers explained the aims and objectives of the focus group discussions to set the scene and the role of both psychologists within the research team. This included how the data generated from the focus group discussions would be used, that is, their goal was to use the findings to further develop the intervention and/or integrate it within usual clinical care to help other people with heart failure.

## Materials

The research team developed a topic guide to structure discussions. Topics included reasons for taking part or declining participation; preference for completing a home-based programme; information required to inform decision-making regarding participation; support requirements to adhere to and complete the programme and potential barriers; and expectations of the programme. All questions were open-ended and prompts were used to generate a deeper understanding of participants' views. Each focus group discussion was audio-recorded and transcribed verbatim.

## Methodological quality and reporting

The research was conducted in accordance with the Consolidated criteria for Reporting Qualitative research to enhance transparency and maximise methodological quality.[15] To reduce bias from responders, members of the team delivering the intervention or involved in the pilot study clinic visits were not involved with focus group discussions.

## Data analysis

Data were analysed by hand using thematic analysis.[16] To maximise methodological quality and trustworthiness of the findings, the following analysis procedures were undertaken: (1) two researchers independently read and reread focus group transcripts (NCO and LA); (2) both researchers independently applied codes to segments of data within the first focus group transcript to develop initial themes and subthemes; (3) the same two researchers discussed findings and agreed a preliminary set of themes and subthemes; (4) one researcher (NCO) repeated this process with the remaining two transcripts; and (5) both researchers agreed the final set of themes and subthemes that adequately represented the data set following discussion. Supporting direct quotes from participants were subsequently applied to themes and subthemes. A third researcher (NO'B) reviewed the themes and subthemes generated to verify interpretation. We did not consider it necessary to use qualitative data analysis software due to the small number of focus group transcripts. The aim was to avoid deterministic processes, which can be a risk with qualitative software, and to prioritise gaining an indepth understanding of the data generated. We achieved this with indepth scrutiny of the transcripts by hand.

## Patient and public involvement

There was no patient and public involvement in the design or planning of the study.

## RESULTS

Three focus groups were conducted, with a duration of 47, 53 and 83 min, respectively. A total of 16 individuals (13 men, 3 women; median age 68 years; IQR: 6.5 years) from the 20 who were eligible to take part in the pilot study and 2 who were eligible but declined participation took part in a focus group discussion. Participants had either completed Active-at-Home-HF (4 participants), were still participating at the time of the focus group (10 participants) or had declined participation (2 participants). Three participants had previously declined centre-based cardiac rehabilitation. Ten participants had implanted cardiac defibrillators. Of the 16 participants, 15 were retired from active work. Following thematic analyses, 10 main themes were generated. It was noted following the conduct of focus group 3 that no new data were generated, that is, it was agreed that data saturation was reached. Table 2 provides a summary of the themes and subthemes, and examples of direct quotes to provide context.

## Fear of undertaking physical activity

Once diagnosed with heart failure, participants reported difficulties overcoming the fear associated with the diagnosis (ie, future cardiac events and overexertion), and increasing physical activity exacerbated this fear. While some reported that they could still engage in some physical activity, there was concern that too much would be detrimental. This was reported to be a significant barrier to participation: "It's frightening when you've gone too far. Something's always holding me back from going that

**Table 2**  Summary of themes and subthemes derived from thematic analyses of focus group discussion transcripts

| Theme | Subtheme | Quotes |
|---|---|---|
| 1. Fear of undertaking physical activity. | | "I'm not confident when I go out on my own. I'm frightened I'm going to collapse." (FG 1, male, aged 57)<br>"It's frightening when you've gone too far. Something's always holding me back from going that little bit further, you know." (FG 1, male, aged 56) |
| 2. Family members influence physical activity efforts. | 2.1. Fear in family members prevents engaging with physical activity. | "I've got a wife who's not happy if I say I'm going out for a walk for an hour…, because she thinks that's too much." (FG 2, male, aged 67)<br>"…Sometimes, your family don't want you to do it, not because they're being horrible, it's because they're concerned about you." (FG 1, male, aged 78) |
| | 2.2. Support from family members facilitates physical activity. | "I needed motivation which I got from the family, which is great…I don't know how I would have fared without family." (FG 2, female, aged 62)<br>"I think it does help if you've got a supportive partner. My husband has come out with me for all my walks right from the start. I think it's good if your partners can be involved and then they realise, don't they, how they can help." (FG 2, female, aged 62) |
| 3. Physical activity programmes require endorsement by clinicians. | | "I think my biggest breakthrough was…I was doing more and more but, again, family was saying, 'Take it easy.' Fortunately, my wife was with me on one occasion when I saw Dr 'XX' and he asked what I'd been doing, and he said, 'Well, do more.' I said, 'How do you mean by that?' He said, 'Just do more and more until you feel you can't and then back off a bit.' So, fortunately, my wife was there and she heard him say it. So, I've had no peace since then." (FG 1, male, aged 78)<br>"My nurse certainly was interested [in my physical activity habits] and when I went last time, was aware of what I was doing and obviously encouraging." (FG 2, female, aged 62) |
| 4. Completion of a cardiac rehabilitation programme provides confidence to complete a physical activity programme at home. | | "I think the most important thing when you've had a heart attack is to have the confidence to do it at all. I think the cardiac rehabilitation that I had gave me the confidence and then you can move on to do other things. I think if you had just had a heart attack and someone says, 'Right, you've got to go for a walk every day,' it would be difficult to do. I think it would. So it's like a step in the process." (FG 2, male, aged 71)<br>"I was really upset when it [cardiac rehabilitation] finished. I had somewhere to go. I'm sat at home by myself all day, so coming here on the Wednesday [for the study visit] and receiving feedback, it was like Christmas all over again, you know. It was great." (FG 1, male, aged 56) |
| 5. Coach support increases motivation long term. | | "[having your coach call you] encourages you to keep up with it, definitely, because I think if you didn't get the phone calls, I think you might just go, 'It's not very nice out today, I'll not bother going.'" (FG 2, female, aged 62)<br>"When somebody is monitoring you, it makes you get up and go out, doesn't it? I mean if you go to the gym yourself, some mornings you might say, 'I'm not going to go today,' but it's helping you, isn't it?" (FG 3, male, aged 62) |
| 6. Weekly agreed targets increased confidence and motivation. | | "It's a target as everyone's saying and you want to do it and you feel very enthusiastic about doing it and I will certainly continue after the 12 weeks because it would be pointless stopping all together wouldn't it? It would waste the benefit sort of thing." (FG 2, male, aged 70)<br>"I think if you were told increase to 12 000 and I come back in 12 weeks' time and we talk about it, you would go…but what's good is on Tuesday [my coach] rings me and goes through all the information, we get an average and then he'll ask me how I'm feeling, etc., etc., and say, 'Right, let's try and take that 12 000 up to 12 500.' That's on a weekly basis. You know for a fact that someone's interested in what's been happening for the past week and we can take that from where we are now to try and improve things." (FG 2, male, aged 69) |

Continued

**Table 2** Continued

| Theme | Subtheme | Quotes |
|-------|----------|--------|
| 7. A credible team increased the likelihood of participation. | | "I think the fact that people like yourselves who are specialists in this sort of area take such an interest in us people. I think that gives you the boost again. I think it does boost people when you've got people who are really even higher than your GP and what not in that specific area of cardiac problems." (FG 2, male, aged 78)<br><br>"Well I think this is a speciality subject, what you're doing and all the stuff here that we've been doing. I think it's better in that situation to get the right advice." (FG 2, male, aged 67) |
| 8. The surrounding environment creates barriers to increased physical activity/exercise. | | "I think another…difficulty…trying to get people motivated is the area that they live in. Now if you've got a bit of countryside, open fields and that, it opens you to more space unlike just got solid concrete, trying to get people motivated to walk down the same street or go a particular…way, that's going to be difficult I think. If you're going to motivate people you're going to have to think where they actually live because people need different types of motivation. For people who live near the coast and the countryside, people living smack bang in the middle of a built-up area etc." (FG 2, male, aged 67)<br><br>"To me, I'm okay most of the time on the flat, but it's any incline. The littlest incline in the world kills me." (FG 2, male, aged 78)<br><br>"I think some of the benefit was the climate as well, it's a struggle in wind and rain." (FG 3, female, aged 63) |
| 9. Participation prompts an increase in everyday activity levels. | | "What I do now is I have to drive to the supermarket but I park in the furthest corner of the car park and I walk round the car park. Then on rainy days, what I have been doing is going into the supermarket and going round it twice before I start my shopping. The people must think you're mad if they look at you on the CCTV." (FG 2, male, aged 71)<br><br>"I think it's got to work doing exercise. I mean they brought me in here; they're going on about putting a defibrillator in. Now when he saw me, how my heart had changed just through doing things, he decided he wouldn't put one in." (FG 3, male, aged 62)<br><br>"I haven't been back to see my specialist at the Freeman [hospital], because he doesn't want to see me for 12 months because I'm too fit for him. I said, 'Are you sick of me?' he said, 'No.' It was 6 months before and then I went for the last check sometime last year, he said, 'Right, I don't want to see you for 12 months. I said, have you gone off me?" (FG 2, male, aged 71)<br><br>"Well since I've completed the course (Active-at-Home-HF), I've been diabetic for 30 years, my blood sugar levels have never been as normal as an ordinary person's in my life." (FG 2, male, aged 79) |
| 10. Support to maintain long-term activity levels would be beneficial. | 10.1. Support from a healthcare professional following completion of the programme would help maintain increased physical activity. | "I think a follow-up is a good idea. You need some sort of follow-up after you've finished. How they do it, whether it's a phone call or a meeting with your doctor or whatever, your GP or anybody like that, I don't know how they would do it. But I think that's quite important that. Then you wouldn't feel as if you've been chucked on the scrap heap type of thing, you've finished, it's done." (FG 3, male, aged 73)<br><br>"Do you know, that's the worst thing, when you've finished your course and the following Wednesday there's no phone call. It's horrendous isn't it?" (FG 2, male, aged 67) |
| | 10.2. Group peer support would promote long-term physical activity. | "I know it's just the start of this programme…, As far as feedback on it is concerned, maybe something about halfway through, maybe six weeks you could have a little meeting with some of the people just for half an hour, just have a little chat and see who's there, what's what, talk to people, something like that anyway. A little informal meeting or a social evening or whatever you want to call it. It lets everybody else know that it's not just you or another two or three people, it might be 20 people." (FG 3, male, aged 73) |

FG, focus group; GP, general practitioner.

little bit further, you know" (focus group (FG) 1, male, aged 56). In this regard, it was reported that reassurance from a clinician is required to confirm that increasing physical activity is safe, and specific information in terms of how much activity should be undertaken to be beneficial, and not detrimental.

### Family members influence engagement with physical activity

It was reported that family members were both a barrier and a facilitator to engaging in physical activity. Some participants reported that family members dissuaded them from engaging in physical activity, although there was a consensus that this was linked to fear and concern: "Sometimes, your family don't want you to do it, not because they're being horrible, it's because they're concerned about you" (FG 2, male, aged 78). Fear of family members often meant that participants were less active following a diagnosis of heart failure than they were previously.

It was also reported that family members were a strong motivating factor once they were reassured that increasing physical activity was beneficial. For some participants, having a partner or relative to support them and participate in new activity routines was reported as a significant facilitator to continued participation. A typical quote included: "I think it does help if you've got a supportive partner. My husband has come out with me for all my walks right from the start. I think it's good if your partner can be involved and then they realise, don't they, how they can help" (FG 2, female, aged 62). There was a consensus that without the practical and emotional support of partners and relatives, increasing physical activity would be more difficult.

### Physical activity programmes require endorsement by clinicians

Endorsement of the physical activity programme by a clinician was reported to reduce patients' fears about increasing levels of physical activity and potentially making their condition worse. Furthermore, it reassured family members when they witnessed clinicians explaining the benefits and advising an increase. In addition, ongoing positive reinforcement by the clinical team was considered paramount to continued participation: "My nurse certainly was interested [in my physical activity levels] and when I went last time, was aware of what I was doing and was encouraging" (FG 2, female, aged 62).

### Prior completion of a cardiac rehabilitation programme provides confidence to engage with home-based physical activity programmes

Cardiac rehabilitation was reported to play a significant role in confidence building among patients with heart failure, including confidence to continue with activities of daily living. As such it was reported to be a significant facilitator to uptake of Active-At-Home-HF as a means of providing patients with ongoing support and encouragement to be physically active in the long term. Continued

participation and feedback was particularly important to those who had completed a cardiac rehabilitation programme: "I was really upset when it [cardiac rehabilitation] finished. I had somewhere to go. I'm sat at home by myself all day, so coming here on the Wednesday [for the study visit] and receive feedback, it was like Christmas all over again, you know. It was great" (FG 1, male, aged 56). This quote highlights the importance of social support and feedback in order to promote uptake and adherence to cardiac rehabilitation. Both were provided via Active-at-Home-HF and were reported to be essential.

### Behavioural support provided by a trained professional increases motivation and adherence

Participants reported that having a trained professional who monitored their performance and provided personalised advice and support promoted continued participation (ie, adherence): "[having your coach call you] encourages you to keep up with it, definitely, because I think if you didn't get the phone calls, I think you might just go, 'It's not very nice out today, I'll not bother going'" (FG 2, female, aged 62). It was important that the support was provided by someone who could advise on the type, amount and duration of activity required to reach a specific target, and someone who was skilled in the use of behavioural strategies who could appropriately challenge participants and support them to overcome perceived barriers.

### Agreed weekly targets increased confidence and motivation

Physical activity goal setting was considered an important mechanism to increase motivation, particularly when weekly goals were personalised and agreed between the participant and the coach. This provided reassurance that goals were realistic, attainable and safe, and it was reported to be rewarding when goals were achieved. Some participants were keen to set themselves daily or weekly targets once they had knowledge and confidence in their capabilities, highlighting the potential long-term impact of the programme.

> It's a target as everyone's saying and you want to do it and you feel very enthusiastic about doing it, and I will certainly continue after the 12 weeks because it would be pointless stopping all together wouldn't it? (FG 2, male, aged 70)

### A credible team increases the likelihood of participation

An incentive of taking part in Active-at-Home-HF was reported to be the support and monitoring offered by experts in heart failure and physical activity. Participants emphasised that endorsement of the programme by a consultant who was knowledgeable about the benefits of physical activity in the context of heart failure and a referral to a trained professional/member of his or her team who was aware of their clinical circumstances helped overcome barriers to ongoing participation. The 'knowledgeable team' provided reassurance around

safety and was skilled in tailoring the intervention to individual needs and preferences, and this was considered important to facilitate initial engagement and adherence.

> Well I think this is a speciality subject, what you're doing and all the staff here know what we've been doing. I think it's better in that situation to get the right advice. (FG 2, male, aged 67)

### The surrounding environment creates barriers to increasing physical activity and maintaining a physically active lifestyle

Environmental factors including living in hilly areas or in surroundings that were not considered picturesque were reported to be barriers to engagement and ongoing participation in physical activity. Poor weather conditions were most frequently reported to impact negatively on motivation and long-term participation.

> To me, I'm okay most of the time on the flat, but it's any incline. The littlest incline in the world kills me. (FG 2, male, aged 78)

> I think some of the benefit was the climate as well, because it was through the summer. (FG 3, female, aged 63)

Participants reported that the support they received via Active-at-Home-HF enabled them to identify ways in which they could increase their levels of activity despite these barriers, for example planning alternative ways to increase their step count and reach their overall goals. In this regard the programme increased motivation and realisation that there was an alternative approach to increasing activity to a level sufficient to have a clinically meaningful impact.

### Participation prompts an increase in everyday activity levels and leads to unexpected positive outcomes

Active-at-Home-HF was reported to encourage participants to seek ways to increase their activity levels by modifying their everyday behaviours (eg, walking instead of driving). One participant reported how the intervention prompted him to spend more time with his grandchildren, thereby positively affecting activity levels and social connections with his family. Some described how their fitness and feeling of well-being had improved (this was confirmed in the pilot study), and the frequency of appointments with their consultant had reduced following progress.

> I haven't been back to see my specialist at the hospital, because he doesn't want to see me for 12 months because I'm too fit for him. I said, 'Are you sick of me?' he said, 'No.' It was six months before and then I went for the last check sometime last year, he said, 'Right, I don't want to see you for 12 months.' (FG 2, male, aged 71)

The intervention led to improvements in cardiac function for the majority of those who participated.[9] For some, this outcome meant avoiding invasive procedures that were planned prior to the study. For other participants the intervention helped to improve cardiac function and control other comorbidities.[9]

> Well since I've completed the course [Active-at-Home-HF], I've been diabetic for 30 years, my blood sugar levels have never been as normal as an ordinary person's in my life. (FG 2, male, aged 79)

### Ongoing support would facilitate long-term maintenance of physical activity

Continued support was considered critically important for psychological and emotional well-being as well as programme adherence and maintenance of activity levels following completion of the intervention. It was emphasised that support should include a review of progress to promote ongoing motivation.

> Do you know, that's the worst thing, when you've finished your course and the following Wednesday there's no phone call. It's horrendous isn't it? (FG 2, male, aged 67)

There was a general consensus that having meetings with others who have taken part in the programme would be beneficial for social support and promoting increased physical activity levels long term.

> Maybe something about halfway through, maybe six weeks you could have a little meeting with some of the people just for half an hour, just have a little chat and see who's there, what's what, talk to people, something like that anyway. A little informal meeting or a social evening or whatever you want to call it. It lets everybody else know that it's not just you or another two or three people, it might be 20 people. (FG 3, male, aged 73)

A consistent theme across focus group discussions was the need for ongoing support to maintain physical activity levels and the clinical benefit observed. Participants reported the omission of this from other programmes they had taken part in, including cardiac rehabilitation. It was acknowledged that provision of support was not necessarily the role of the consultant; however, it was important to have a trained team member who was motivated to support patients to increase and maintain their activity levels and who was knowledgeable in this area. Peer support was considered important for long-term maintenance of physical activity following completion of the programme as a means of continuing activity consistently and safely.

### DISCUSSION

Although physical activity is widely advocated to improve health, it is viewed as a significant challenge for people living with chronic health conditions and regularly increases fear. Fear of increasing physical activity was reported in the present study, and as such the need for

positive reinforcement and reassurance from clinicians who are advocates of physical activity in the context of heart failure was considered necessary to overcome this barrier. Family members and partners were reported to reinforce fear and restricted activity levels of participants for fear of an adverse event occurring. Involvement of significant others in consultations could provide a solution to this problem and should be explored further.

Participating in cardiac rehabilitation was reported to reduce psychological stress and fear associated with physical activity and was a facilitator to uptake of Active-at-Home-HF. This finding was further substantiated by the sample of participants taking part in the study (ie, it included those who had previously completed centre-based cardiac rehabilitation who were keen to continue). Just like community-based or clinic-based cardiac rehabilitation, it is important for home-based interventions to be holistic where possible.[17] Duda and colleagues[18] reported that the success of an exercise intervention depended on the intensity and type of support offered. Active-at-Home-HF involved the use of lifestyle coaches who were trained to use specific evidence-based behaviour change techniques to target increased physical activity, and this training was essential to provide the necessary skills and confidence to effectively deliver the intervention. Although coaches, through their academic training, were able to communicate the physiological and health benefits of increased physical activity in the context of heart failure, which is important for motivation/engagement[19] and adherence,[20] they reported that without behavioural training they would have struggled specifically to support participants to overcome barriers and this would have impacted on adherence. This suggests that training of those required to deliver the intervention is critically important to support enactment and maintenance of physical activity. Therefore, behavioural training and the provision of a standardised proforma were considered facilitators to intervention delivery.

The need to involve clinicians to advocate and support increased physical activity at home was considered vital to reassure patients about safety and efficacy, to add credibility and to successfully integrate the programme into routine clinical care. In this regard participants were more likely to participate and adhere to the programme if a clinician had advocated it and were made aware of their progress. A potential limitation of our study is that only two individuals who declined participation in Active-at-Home-HF took part in a focus group discussion. This provides limited insights in terms of barriers to uptake and this should be explored further, although encouragingly 25% of our sample had previously declined participation in centre-based cardiac rehabilitation programmes. Our findings suggest that the decision to participate in Active-at-Home-HF was largely due to the way in which it was offered to patients by clinicians who were positive about the programme and who could provide reassurance about its safety and overall benefits. This is further reflected in our sample, which included participants who had been

diagnosed with heart failure for an average of 10 years (SD±9 years), which could be considered unusual (ie, it may have been expected that those newly diagnosed would be more likely to engage). However, our sample included those who had previously declined cardiac rehabilitation, those who had attended and those who had a long history of heart failure who had never been offered rehabilitation, demonstrating the flexibility of the intervention and its acceptance to a broad range of patients.

Currently, attendance at cardiac rehabilitation in hospitals or community centres involves frequent travel and associated costs, and these are barriers to participation, especially for those who have limited financial resources or who are unable to drive. Furthermore, for some patients a group-based environment reduces confidence and does not always provide the one-to-one support they require. Active-at-Home-HF is an alternative to help overcome these barriers while providing patients with a personalised programme capable of increasing activity levels and impacting positively on health and fitness.

Previous studies support the use of supportive calls as an effective way to motivate patients to maintain physical activity levels, particularly those who have a low level of physical functioning.[21] The Active-at-Home-HF intervention was delivered by short telephone calls and this proved to be an effective way of targeting and maintaining increased physical activity while providing support. The brief nature of these telephone interactions increases the likelihood that this intervention could be feasibly rolled out within routine clinical practice. However, our training programme focused only on the delivery of behavioural strategies by coaches who were already knowledgeable about the role of physical activity in the context of heart failure. Therefore, this suggests the potential need for training developed specifically for clinicians to ensure they have the knowledge and skills required to effectively deliver all aspects of the intervention. A commonly suggested facilitator to ongoing engagement and maintenance of physical activity levels following completion of the programme was having an opportunity to meet with others who had completed the programme as a means of social support. This is an important consideration, with 40%–50% of those participating in physical activity programmes relapsing and returning to their previous physical activity states within 6 months.[21 22] Such groups provide a strong form of social support where long-term friendships are developed that are essential for long-term engagement and encouragement.[20] Furthermore, these groups bring together individuals with similar conditions and work to generate a sense of obligation, thereby facilitating adherence, and this was highlighted by participants.[23] It is important to highlight that group support was suggested mainly at the postintervention time-point to promote maintenance. This reinforces the need for some to gain mastery and confidence around increasing their activity levels before entering a group environment.

To the best of our knowledge, this is the first qualitative study to explore barriers and facilitators to

participation and adherence to a home-based physical activity programme in the context of heart failure. In addition, we identified ways in which the intervention could be successfully integrated within routine clinical care. Our findings highlight that facilitators to home-based physical activity programmes include endorsement by members of the clinical team, where possible a consultant, and that clinicians should be advocates of and knowledgeable about physical activity in the context of heart failure. While it is acknowledged that not all clinical teams will have the knowledge, expertise and skills to effectively promote physical activity, the findings from our study suggest that a future intervention should incorporate training for clinicians to address this issue and this would help facilitate future roll-out. Those who declined participation in Active-at-Home-HF did so out of fear and lack of knowledge about the specific benefits of physical activity in the context of heart failure. Specifically, participants believed that physical activity would not improve their condition, and fear of a cardiac event outweighed any possible benefits of an intervention. Findings also suggested that partners/family members reinforced fear and beliefs about efficacy. This suggests that a future intervention should incorporate specific learning/education content about the role of physical activity in addition to a recommendation from a clinical team member, and involve family members, where possible, to overcome misconceptions. Participants were supported throughout the duration of the intervention by research team members with background in clinical exercise physiology who were trained by a health psychologist to use evidence-based behaviour change techniques to target increased physical activity and maintenance over time. Participants reported this support as essential to obtain feedback, maintain motivation and to challenge their perceptions, and it was beneficial as a form of social support. Therefore a future intervention should involve training of clinical team members to provide this support to facilitate full integration of the intervention in routine clinical care.

In summary, the most salient facilitators to uptake and engagement of the Active-at-Home-HF intervention were clinical team members who provided personalised information about the benefits and safety of the intervention, and involvement of family members during consultations to provide them with reassurance and encourage them to provide support. In the context of health behaviour change theory, this approach targets attitudes, beliefs, risk perceptions and self-efficacy of patients that increases the likelihood of intention formation (ie, engagement with the programme). Facilitators to adherence included behavioural prompting (ie, a telephone call) by a knowledgeable team member who can provide individualised feedback and support to enable patients to set and revise activity goals and identify ways to overcome barriers to reaching these goals. In combination these self-regulatory strategies target adherence and support maintenance of physical activity. Positive reinforcement from a consultant (eg, positive feedback on increased activity levels in relation to outcomes) was considered a further facilitator to adherence. Although involvement of family members was not a formal component of the Active-at-Home intervention, in instances where participants did involve family members in their consultations, they reported this to be of significant benefit to adherence. Future delivery of an optimised version of this intervention should consider uptake, engagement and adherence separately and use the facilitators identified to target each.

Although it is possible that patients referred to rehabilitation with other long-term health conditions can present with unique challenges that create barriers to uptake, engagement and adherence, the findings of our qualitative study revealed parallels with other clinical groups. A 2018 systematic review of qualitative studies in the context of chronic obstructive pulmonary disease reported fear and lack of self-efficacy as barriers to uptake and engagement, and the relationship with healthcare professionals, personalised feedback and peer interaction as facilitators to engagement and adherence.[24] Similarly in the context of stroke, barriers and facilitators included fear of a subsequent stroke, and personalised feedback and support to promote adherence.[25] These findings suggest that commonalities exist, and the way in which we communicate and support patients to increase and maintain physical activity levels is vitally important. As such there is potential to adapt the Active-at-Home-HF intervention for delivery in a range of contexts.

## Author affiliations

[1]Translational and Clinical Research Institute, Newcastle University Faculty of Medical Sciences, Newcastle upon Tyne, United Kingdom
[2]Department of Psychology, Northumbria University - City Campus, Newcastle upon Tyne, United Kingdom
[3]School of Health and Social Sciences, Buckinghamshire New University, High Wycombe, United Kingdom
[4]Cardiology, Newcastle upon Tyne Hospitals NHS Foundation Trust, Newcastle upon Tyne, United Kingdom
[5]Biosciences Institute, Newcastle University, Newcastle upon Tyne, United Kingdom
[6]Faculty of Health and Life Scienes, Coventry University, Coventry, United Kingdom
[7]School of Health & Life Sciences, Teesside University, Middlesbrough, United Kingdom

**Acknowledgements** The authors would like to acknowledge and thank the participants who took part in this study.

**Contributors** LA, DGJ and NCO conceived the study and its design. LA supervised its conduct. KB and GAM facilitated recruitment of participants and collected demographic data. SJC, SC and NCO delivered the Active-at-Home-HF intervention and conducted study visits that involved data collection. LA and NO'B conducted all focus group discussions. NCO, LA and NO'B analysed and interpreted the data. NCO and LA drafted the manuscript. LA, NCO, DGJ, DB, KB, GAM, SJC, SC and NO'B made an important intellectual contribution, revised the manuscript and approved the version to be published. LA is accountable for all aspects of the study, including the accuracy and integrity of the data.

**Funding** This research was funded by the National Institute for Health Research Biomedical Research Centre (grant number: BH142109).

**Competing interests** None declared.

**Patient and public involvement** Patients and/or the public were not involved in the design, or conduct, or reporting, or dissemination plans of this research.

**Patient consent for publication** Not required.

**Ethics approval** This study was approved by the NHS North East - Tyne & Wear South Research Ethics Committee (REC reference: 15/NE/0190).

**Provenance and peer review** Not commissioned; externally peer reviewed.

**Data availability statement** No data are available. Participants were not asked to provide consent to share their transcripts beyond the research team.

**ORCID iDs**
Guy A MacGowan http://orcid.org/0000-0002-2685-4165
Leah Avery http://orcid.org/0000-0003-3578-1209

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
