## [Reviewer comments · BMJ Open]

ARTICLE DETAILS

TITLE (PROVISIONAL)	Overcoming barriers to engagement and adherence to a home-based physical activity intervention for patients with heart failure: A qualitative focus group study
AUTHORS	Okwose, Nduka; O'Brien, Nicola; Charman, Sarah; Cassidy, Sophie; Brodie, David; Bailey, Kristian; MacGowan, Guy; Jakovljevic, Djordje; Avery, Leah

VERSION 1 – REVIEW

REVIEWER	Patrick Phillips Sheffield Teaching Hospitals NHS Trust United Kingdom
REVIEW RETURNED	06-Mar-2020

GENERAL COMMENTS	Thank you for the opportunity to review this well written and interesting paper, I have some comments and questions and I think addressing these might be beneficial, however the paper is of good overall quality. General comments below followed by more specific comments line by line. I'm curious about the relationship between this qualitative study, the pilot study and the development process of the intervention. Superficially the paper is presented as an investigation into barriers to engagement and attendance at home based HF rehab - but specifically it is an investigation into barriers and facilitators related to a particular intervention - is this study explicitly part of the development of the active at home HF intervention? Additionally I wonder about the impact of participation in the pilot study and the research process on the sample and their experiences of the intervention and associated barriers and facilitators - it appears that all participants were involved in the pilot study and there are limited numbers of non-participants included in the focus groups (two, I think) so the views expressed are perhaps overwhelmingly those of people engaged enough to participate in both the HF programme and the associated research - does this affect your findings? You don't seem to have directly referenced any published papers detailing the development or piloting of the active at home intervention and I think if these are available they should be referenced. I'd also be interested to know what the local model of cardiac rehabilitation is for these patients - is the active at home HF
--

programme intended as a follow on or alternative to a more formal programme it seems like quite a number had previously taken part in CR programmes.

A couple of things that I think could be addressed in slightly more depth in the discussion are:

1/ a contrast of the barriers and facilitators associated with the different issues of uptake and adherence

2/ I think you could devote part of the paper specifically to the findings related to the non-participants.

page 2 line 43 - all participants had heart failure for 9 years or more?

page 2 line 51 - reference to active ingredients of the intervention suggest that the study is specific to the active at home programme

page 3 line 35 - I think only having 2 non-participants is a limitation and should be discussed

page 4 line 8 - it's not clear in this discussion (attendance at cardiac rehab) which groups you are talking about - 45% of what population...?

page 4 line 22 typo have shown

page 5 line 8 - the referenced papers don't seem relevant to this statement - 2 citations are to methodology papers

page 5 line 37 - is this the correct reference

Page 5 line - I would like to see a table with more detail of participant characteristics

page 6 line 20 - recruitment and selection is not totally clear - were all FG participants also participants in a pilot study of the active at home intervention

page 6 line 23 - were the psychologists running the focus groups involved in the development of the active at home intervention or totally independent of the whole project?

page 8 line 18 - any use of software for data analysis?

page 8 line 51 - some clarity would be beneficial here re: where these twenty pre screened and identified as meeting the requirements of your purposive sample - or do these 20 represent a consecutive, convenience sample who were then weaned down to 16?

Table 1 - there is a lot of duplication between this table and the text - more of an editorial decision but maybe worth considering reducing the amount of information in the table? Perhaps removing the direct quotations?

page 16 line 58 - a little confused about the reference to coming here on a Wednesday - I thought it was home based?

page 19 line 56 and page 20 line 4 - the statement that the intervention led to improvement in cardiac function is not supported by any evidence and it should be stated that this is a perception of the participants.

COREQ statement - there are quite a few N/A responses - I think some of these should be addressed, for instance - is there a description of minor themes/ diverse cases, non-participants are a case in point that could be discussed explicitly. I think it would also help transparency to explicitly state whether field notes were taken, or software used, or participant feedback obtained etc.

Best Wishes

REVIEWER	Scott Kehler Dalhousie University, Canada
REVIEW RETURNED	15-Mar-2020

GENERAL COMMENTS	Thank you for the opportunity to review this manuscript by Okwose et al. which qualitatively examined barriers and facilitators to adhering in a home-based physical activity program amongst 16 patients diagnosed with heart failure for 9+ years. Here, the authors determined that the major barrier to PA participation was fear. Families were either barriers (fear) or facilitators (providing social support). Strong factors related to PA participation were healthcare provider endorsement, continued support and feedback from trained personnel, and personalized approaches to overcoming barriers and increasing confidence. This is a well-written and informative document. I have few suggestions for this manuscript. Can the authors comment on what might be expected responses re: barriers and facilitators from those delivering the home-based program? The patient perspective is very useful, but it would also be important to know from the delivery perspective How might the results here inform future home-based cardiac rehabilitation programs? It was mentioned that a recurring theme was ongoing support as a strong facilitator. How could continued support be feasibly rolled out? The authors go some way to comment on this, but could they please expand on their current discussion? The authors indicate that this is the first qualitative study to examine barriers and facilitators to a home-based rehab program in patients with heart failure. Can the authors please comment on some unique challenges/barriers in patients with heart failure vs. other patients who would be referred to a rehab program?
---

VERSION 1 – AUTHOR RESPONSE

Reviewer: 1

1. Thank you for the opportunity to review this well written and interesting paper, I have some comments and questions and I think addressing these might be beneficial, however the paper is of good overall quality.

Thank you for your positive comments about our manuscript, and for taking the time to provide a review.

2. I'm curious about the relationship between this qualitative study, the pilot study and the development process of the intervention. Superficially the paper is presented as an investigation into barriers to engagement and attendance at home based HF rehab - but specifically it is an investigation into barriers and facilitators related to a particular intervention - is this study explicitly part of the development of the active at home HF intervention?

This study aimed to identify barriers and facilitators to engagement and adherence to the Active-at-Home-HF intervention and to determine whether it could be delivered as part of routine clinical care. However, the findings have provided clear direction in terms of how the intervention should be re-developed for delivery by clinicians in the future. This is an important point to communicate, therefore we have inserted additional information in to the manuscript for clarity (Page 4, paragraph 3, lines 17-18) and provided additional information throughout, where necessary to reiterate this important point.

3. Additionally I wonder about the impact of participation in the pilot study and the research process on the sample and their experiences of the intervention and associated barriers and facilitators - it appears that all participants were involved in the pilot study and there are limited numbers of non-participants included in the focus groups (two, I think) so the views expressed are perhaps overwhelmingly those of people engaged enough to participate in both the HF programme and the associated research - does this affect your findings?

Thank you, this is an interesting and important point. One of the main findings of our qualitative study helps to respond to this comment we hope. Active-at-Home-HF was endorsed by consultants who were knowledgeable about the programme and advocates of physical activity in the context of heart failure. Participants reported this as being vitally important to provide reassurance about safety and efficacy. While the sample of participants included those who had participated in cardiac rehab previously, it also included participants who had declined centre-based cardiac rehab. We hadn't explicitly reported this information previously, therefore we have included it in our revised manuscript (Page 9, paragraph 3 [results]). We acknowledge the point made about the sample being made up of those who are already engaged, however we believe the endorsement of clinicians was an important contributing factor to uptake and engagement and is important for other clinical teams should they wish to use the intervention. We have further emphasised this point in our discussion (Page 24, paragraph 2, lines 5-7).

4. You don't seem to have directly referenced any published papers detailing the development or piloting of the active at home intervention and I think if these are available they should be referenced.

Thank you for highlighting this. At the time of submission our pilot study manuscript was also under review. It has since been published, therefore we have provided a reference (Page 5 [Ref 9]).

5. I'd also be interested to know what the local model of cardiac rehabilitation is for these patients - is the active at home HF programme intended as a follow on or alternative to a more formal programme it seems like quite a number had previously taken part in CR programmes.

Active-at-home-HF was developed both as an alternative to centre-based cardiac rehabilitation and as a means to continue rehabilitation in recognition that many patients express a desire to continue. We have provided additional information to clarify (Page 5, lines 3-7).

6. A couple of things that I think could be addressed in slightly more depth in the discussion are:
1/ a contrast of the barriers and facilitators associated with the different issues of uptake and adherence
2/ I think you could devote part of the paper specifically to the findings related to the non-participants.
Thank you for this helpful suggestion. Please see page 27 where we have provided an additional paragraph to address the first point raised by the reviewer (paragraph 2).
The second point has been addressed on page 26, paragraph 2, Lines 11-18.
7. Page 2 line 43 - all participants had heart failure for 9 years or more?
Thank you for highlighting this error. 9 years was the standard deviation and not the mean, apologies. We have corrected this in our revised manuscript (Page 2: Abstract and presented this information in Table 1).
8. Page 2 line 51 - reference to active ingredients of the intervention suggest that the study is specific to the active at home programme
The study is specific to Active-at-Home-HF, however we explored general barriers and facilitators to uptake, engagement and adherence to home-based intervention. In our experience it is difficult to obtain an in depth account of barriers and facilitators without participants having experienced an intervention they can critique. Therefore, the aim was to provide an intervention and elicit feedback. We have provided additional information for clarity (Page 6, paragraph 1).
9. Page 3 line 35 - I think only having 2 non-participants is a limitation and should be discussed
Thank you for highlighting this point. We agree and have provided additional information in the discussion section of the manuscript (Page 24, paragraph 2 lines 7-12).
10. Page 4 line 8 - it's not clear in this discussion (attendance at cardiac rehab) which groups you are talking about - 45% of what population...?
Apologies. We agree that this is not clear. 45% relates to the percentage of patients following a cardiac event. We have provided additional information for clarity (Page 4, lines 5-6).
11. Page 5 line 8 - the referenced papers don't seem relevant to this statement - 2 citations are to methodology papers
Thank you for highlighting this error. We have checked the references throughout and corrected the errors.
12. Page 5 line 37 - is this the correct reference
We have checked all in-text citations and the end reference list and corrected the errors identified.
Thank you for bringing these to our attention.
13. Page 5 line - I would like to see a table with more detail of participant characteristics
We have added a table to our revised manuscript summarising participant characteristics (Page 6 – Table 1). Thank you for this useful suggestion.
14. Page 6 line 20 - recruitment and selection is not totally clear - were all FG participants also participants in a pilot study of the active at home intervention
All but two participants were participants our pilot study. We have clarified this point (Page 7, lines 5-6).
15. Page 6 line 23 - were the psychologists running the focus groups involved in the development of the active at home intervention or totally independent of the whole project?

One of the two psychologists who conducted the focus group discussions developed the behavioural intervention. We have added this important information to the manuscript (Page 7, paragraph 1, lines 5-6).

16. Page 8 line 18 - any use of software for data analysis?
The focus group transcripts were coded and analysed by hand without the use of software. We have provided justification for this decision in the revised version of the manuscript (Page 9, paragraph 1, Lines 6-10).

17. Page 8 line 51 - some clarity would be beneficial here re: where these twenty pre-screened and identified as meeting the requirements of your purposive sample - or do these 20 represent a consecutive, convenience sample who were then weaned down to 16?
Twenty pre-screened participants took part in the pilot study of the intervention, and 14 of these participants were purposively sampled to take part in a focus group discussion. In addition, two participants who declined to take part in the intervention attended a focus group discussion. We have provided additional information and hope that this adds clarity (Page 9, results section).

18. Table 1 - there is a lot of duplication between this table and the text - more of an editorial decision but maybe worth considering reducing the amount of information in the table? Perhaps removing the direct quotations?
It would be our preference to report the table as it is because it provides a full overview of findings from the data generated. The direct quotes substantiate the themes and provide credibility to increase trustworthiness, a key methodologically quality item. We are willing to amend the table if the Editor feels that is necessary/if it is preferred (NB: Table 1 has now been re-labelled Table 2 in our revised manuscript).

19. Page 16 line 58 - a little confused about the reference to coming here on a Wednesday - I thought it was home based?
Thank you for pointing this out. We have checked this and it was a typographical error. We have now corrected it. (Page 19, line 1).

20. Page 19 line 56 and page 20 line 4 - the statement that the intervention led to improvement in cardiac function is not supported by any evidence and it should be stated that this is a perception of the participants.
Our pilot study reported an improvement in cardiac function for the majority of participants. We have provided a reference to support this statement (Page 22, line 2 and 4).

21. COREQ statement - there are quite a few N/A responses - I think some of these should be addressed, for instance - is there a description of minor themes/ diverse cases, non-participants are a case in point that could be discussed explicitly. I think it would also help transparency to explicitly state whether field notes were taken, or software used, or participant feedback obtained etc.
We have provided additional information for items 7 (information communicated to participants researcher goals), 27 (analysis software) and 32 (reference to minor/sub-themes) (Pages 8, 9 and 11). An updated COREQ statement has been uploaded.

Reviewer: 2

1. This is a well-written and informative document. I have few suggestions for this manuscript. We would like to thank the reviewer for their positive response to our manuscript.
2. Can the authors comment on what might be expected responses re: barriers and facilitators from those delivering the home-based program? The patient perspective is very useful, but it would also be important to know from the delivery perspective.

This is an interesting question, and thank you for raising it. Three members of the research team were trained to deliver the intervention and used a proforma to guide delivery. Following delivery, they agreed that without the training it would have been difficult to support participants to overcome barriers to physical activity specifically. We have added these insights and others to our revised manuscript (Page 24, lines 2-11).

3. How might the results here inform future home-based cardiac rehabilitation programs? It was mentioned that a recurring theme was ongoing support as a strong facilitator. How could continued support be feasibly rolled out? The authors go some way to comment on this, but could they please expand on their current discussion?
Participants reported the usefulness, and in some cases the necessity for support from a trained individual (preferably a healthcare professional) by telephone; from family members at home; and from others who had completed the intervention (e.g., peer support post intervention). Therefore, roll-out relies on engaging with these three groups to inform an optimised intervention. We have provided additional information to add clarity (Page 25, paragraph 2 lines 5-11, Page 26, paragraph 2, lines 4-5, 10 and Page 27, lines 2-3.)
4. The authors indicate that this is the first qualitative study to examine barriers and facilitators to a home-based rehab program in patients with heart failure. Can the authors please comment on some unique challenges/barriers in patients with heart failure vs. other patients who would be referred to a rehab program?
This is another interesting and important question. We have provided a response that we hope answers the question (Page 27-28).

VERSION 2 – REVIEW

REVIEWER	Patrick Phillips Sheffield Teaching Hospitals NHS Foundation Trust United Kingdom
REVIEW RETURNED	17-Apr-2020
GENERAL COMMENTS	Hi, thank-you for your positive responses to my comments on the previous version of this paper - I just have 2 very minor queries/ comments which relate directly to changes made from the initial version: Can you clarify heart failure duration numbers? Numbers quoted suggest that on average participants in your study had lived with heart failure for 10 years and by definition some had lived with it for considerably longer than that (SD +/- 9 years) - average post diagnosis survival is around 5 years in the UK with around 25% surviving to 10 years. I'd anticipate that most people attending cardiac rehabilitation might be those that are relatively newly diagnosed, suggesting that your sample is perhaps unusual?

	Does the direct quotation below suggest that the participant values involvement in the study rather than the home based cardiac rehab? 'I was really upset when it (cardiac rehabilitation) finished. I had somewhere to go. I'm sat at home all day, so coming here on the Wednesday (for the study visit...)'
REVIEWER	Scott Kehler Dalhousie University. Canada
REVIEW RETURNED	21-Apr-2020
GENERAL COMMENTS	The authors have done a great job addressing the reviewer comments.

VERSION 2 – AUTHOR RESPONSE

Reviewer 1:

Thank you for your positive responses to my comments on the previous version of this paper. I just have 2 very minor queries/comments, which relate directly to changes made from the initial version.

Comment 1: Can you clarify heart failure duration numbers? Numbers quoted suggest that on average participants in your study had lived with heart failure for 10 years and by definition some had lived with it for considerably longer than that (SD +/- 9 years) - average post diagnosis survival is around 5 years in the UK with around 25% surviving to 10 years. I'd anticipate that most people attending cardiac rehabilitation might be those that are relatively newly diagnosed, suggesting that your sample is perhaps unusual?

Response 1: We agree that our sample is quite unusual; however, this also suggests that our intervention appeals to a broad range of heart failure patients. Interestingly, it was taken up by those who have previously completed centre-based cardiac rehabilitation, and therefore demonstrates that the intervention is considered an acceptable means of continuing the rehabilitation process. We have highlighted these points (Page 22, paragraph 3 lines 2-5 and Page 24, paragraph 1, lines 1-7).

Comment 2: Does the direct quotation below suggest that the participant values involvement in the study rather than the home based cardiac rehab?

'I was really upset when it (cardiac rehabilitation) finished. I had somewhere to go. I'm sat at home all day, so coming here on the Wednesday (for the study visit...)'

Response 2: This is an interesting question. However, valuing involvement in the study rather than the intervention does not fully explain adherence (i.e., study visits occurred at the beginning and end of the 12-week intervention only). What we believe this quote highlights the need for social support and feedback to support the cardiac rehabilitation process. We have reported this in our revised manuscript (Page 18, paragraph 1 lines 1-3).

Reviewer: 2

The authors have done a great job addressing the reviewer comments.

We would like to thank Reviewer 2 for taking the time to review our revised manuscript.